# Good conduct makes your face attractive: The effect of personality perception on facial attractiveness judgments

**Ryosuke Niimi** *, **Mami Goto**

Faculty of Humanities, Niigata University, Niigata, Japan

* niimi@human.niigata-u.ac.jp

## Abstract

Human facial attractiveness is related to physical features, such as clear complexion and symmetry. However, it is also known that facial attractiveness judgments are influenced by a wide range of non-physical factors. Here, we examined the effect of the personality information of a target person on facial attractiveness judgments. In Experiment 1, participants read a verbal description of a target person (high or low honesty), followed by the presentation of the target face and facial attractiveness rating. The honest personality increased the rated facial attractiveness, replicating a previous report. This "honesty premium" effect was independent of pre-rated facial attractiveness (Experiment 1), target gender, participant gender, and target clothing (Experiment 2). Experiment 3 found that creative personality did not affect facial attractiveness ratings, while an aggressive personality was suggested to decrease the rated facial attractiveness of male targets. We did not find evidence that participants' moods caused these effects. The results suggest that the "what is good is beautiful" stereotype is robust and that facial attractiveness is malleable and dependent on various physical and non-physical information.

## Introduction

Human physical attractiveness has drawn considerable attention from researchers in psychology and related fields. Most people are at least somewhat concerned about their physical appearance, which sometimes affects their psychological wellbeing [1]. Facial attractiveness is a major concern, as it is widely believed that it is related to personality traits and therefore changes the behavior of others. Empirical studies have shown that people have a naïve belief that "what is beautiful is good." A target person with an attractive face is often rated as having a favorable personality, good health, and high competence, which is known as the physical attractiveness stereotype [2–8]. It is noteworthy that the validity of such a stereotype has been questioned, as physical attractiveness does not reliably predict actual competence or personality [5, 9, 10], although some studies suggest a partial relationship between facial appearance and actual personality [11]. Nevertheless, facial and overall physical attractiveness actually influence a wide range of human behaviors, including dating [12, 13], helping [14, 15],

**Data Availability Statement:** All data files are available in the OSF repository (https://doi.org/10.17605/OSF.IO/RYSNM).

**Funding:** This study was supported by a grant from the KOSE Kosmetology Research Foundation

(https://www.kose-cosmetology.or.jp/) awarded to RN (J-20-30). The funder did not play any role in the study design, data collection and analysis, decision to publish, or preparation of the manuscript.

**Competing interests:** The first author (RN) received a grant from the KOSE Kosmetology Research Foundation. This does not alter our adherence to PLOS ONE policies on sharing data and materials. The scond author (MG) declares no competing interests.

persuasion [16], personnel decision [17], and voting [18–20]. Virtual avatars have similar effects [21, 22].

What allows facial attractiveness to have these pervasive influences is the high sensitivity of human vision to attractiveness. Visual perception of facial attractiveness is efficient [23, 24] and consistent among observers [7]. These facts have led researchers to hypothesize that facial attractiveness is a quasi-objective visual feature and that human vision is tuned for it. Indeed, studies have sought objective determinants of facial attractiveness, such as the size of facial parts [25], symmetry [26, 27], and averageness [28], although debate about these factors persist [29, 30]. These findings are consistent with the evolutionary account for attractiveness [31, 32], in which those objective determinants are signals for "good genes" and the human visual system has evolved to detect those signals, although there are arguments against this account [5, 33].

However, it seems quite important to remember that attractiveness is a psychological construct that represents an observer's internal perception of being attracted, not an objective entity: beauty is in the eye of the beholder. Knowing that facial attractiveness is malleable and dependent on many psychological factors in addition to physical ones may provide some insights into how we can cope with appearance-related prejudices and stereotypes that may undermine psychological wellbeing [1].

Perceptions of facial attractiveness are not always based on physical features. They are influenced considerably by idiosyncratic standards of beauty [34]. The observer's experiences also play a crucial role, as the physical appearances of individuals that the observer knows well are often judged as more attractive [35–37]. Literature has reported countless factors affecting perceptions of facial attractiveness, such as familiarity [38–40], observer's mood [41, 42], alcohol consumption [43], and nicotine intake [44]. Accessory stimuli accompanying a target face matter as well, including facial expressions [45, 46], hairstyle [47], flanking faces [48], bowing [49], and odor [50]. Red color was first shown to increase facial attractiveness [51] but a later meta-analysis revealed that the effect is negligible [52].

## The effect of personality perception on facial attractiveness

Prior knowledge about a target person influences rated facial attractiveness. Thiruchselvam et al. [40] presented (ostensible) peer attractiveness ratings before a target face was rated, and participants' ratings of the target's facial attractiveness were biased by the peer ratings. Situational context [53], attitude similarity [54] and in-group bias [38] are also known to influence facial attractiveness judgments. Favorable behaviors of a target person increase attractiveness. For example, using video stimuli, Nisbett and Wilson [55] demonstrated that the physical appearance of a target person behaving warmly and friendly was rated as more appealing than cold and distant targets.

Verbal descriptions of target personalities can also alter attractiveness judgments. Gross and Crofton [56] presented a photograph of a (fictional) female student attached with verbal descriptions of her personality. Male and female participants gave higher physical attractiveness ratings under the condition of favorable descriptions (e.g., generous, friendly) than under the condition of unfavorable descriptions (selfish, unfriendly). Hassin and Trope [57] (Study 6) adopted a similar procedure and found that descriptions of the kind personality of a male target increased his facial attractiveness rated by male and female participants. Such an effect seems to include both an increment effect by positive personality information and a decrement effect by negative personality information [58]. A comparable effect of personality information has been reported for female bodily attractiveness [59]. These findings suggest that there is a "what is good is beautiful" stereotype, the reverse of the physical attractiveness stereotype.

Paunonen [60] replicated and extended these results using a modified experimental procedure. Unlike in the previous studies [56, 57], personality descriptions were presented prior to the presentation of the target face image. This procedure eliminated the possibility that the facial image affected the interpretation of the descriptions. He also extended research of the "good is beautiful" effect by using personality traits not manipulated in the previous studies. Participants read a personality description that included manipulations of honesty, intelligence, and independence, followed by personality ratings (manipulation check). Then, a face image of a target person was shown, and the participants rated its physical characteristics, including attractiveness. The results showed an "honesty premium" effect, i.e., that target descriptions of high honesty increased the rated facial attractiveness compared to low honesty descriptions. Intelligence and independence did not yield such effect.

Later studies [61, 62] further confirmed that the presentation of favorable personality information did increase attractiveness by comparing the attractiveness ratings before and after the personality information presentation. This "what is good is beautiful" effect was found irrespective of the level of pre-rated facial attractiveness [56, 61]. Another intriguing aspect of these studies is the fact that personality information often altered the ratings of physical characteristics other than attractiveness (e.g., kind targets were rated as having shorter ears and rounder chins [57]), further implying that these effects reflect the stereotype that physical features are related to personality traits.

## The present study

We conducted four experiments to replicate and extend the results of Paunonen's study [60] (see Fig 1a). First, we aimed to replicate the "honesty premium" effect on facial attractiveness with a Japanese student sample (Experiment 1) and a larger sample including a wide range of ages (Experiment 2). Furthermore, we examined whether personality traits other than honesty (creativity and aggressiveness) would show similar effects (Experiment 3).

In addition, Experiment 4 examined the effects of mood (see Fig 1a). Paunonen concluded that the "honesty premium" effect was mediated by the general likability of a target person:

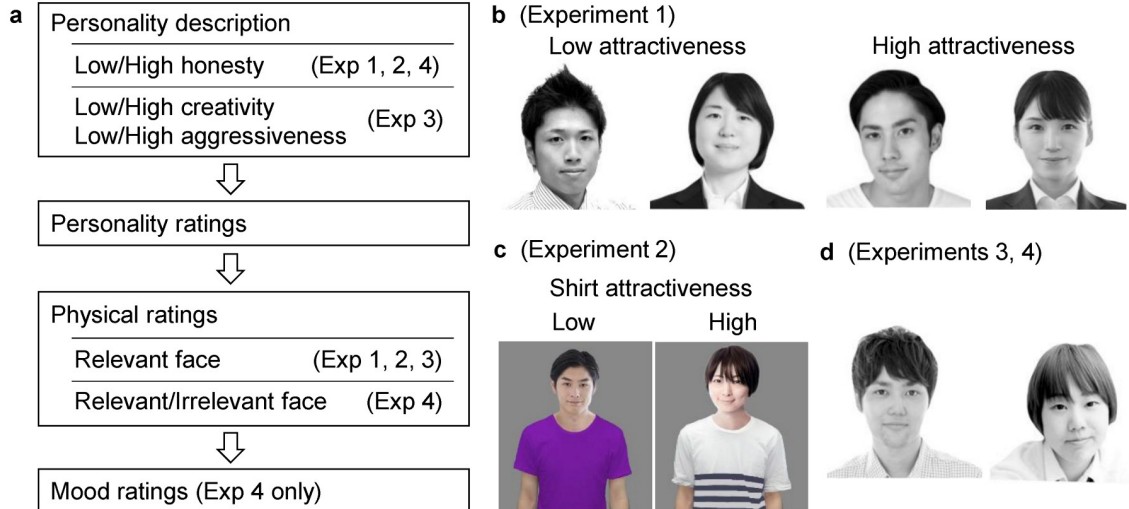

**Fig 1. An overview of the experimental procedure.** Schematic illustrations of the procedures (**a**) and examples of the stimulus face images (**b, c, d**) are shown. Pre-rated facial attractiveness was manipulated in Experiment 1 (**b**). Experiments 2, 3, and 4 adopted the faces of medium attractiveness (**c, d**). Shirt attractiveness was varied in Experiment 2 to test whether it influenced the facial attractiveness ratings (**c**).

rated likability was higher for the honest target and covaried with rated attractiveness. However, reading the honest personality description might have induced a positive mood in the participants, while the low honesty description might have induced a negative mood. It is known that positive/negative moods influence face perception [63], trustworthiness and friendliness of faces [64], and facial attractiveness [41, 42] (see also [65]). In these studies, moods were induced in ways that were irrelevant to the main tasks of the facial attractiveness rating (e.g., background music), so it seemed probable that the "honesty premium" effect of Paunonen [60] might be due to the effect of positive/negative mood induced by the high/low honesty descriptions (the mood hypothesis).

## Experiment 1

Experiment 1 replicated Paunonen's [60] experiment with some exceptions. We manipulated the honesty of the personality description but did not manipulate intelligence or independence. Additionally, we manipulated the pre-rated attractiveness of the stimulus face images. As studies have shown that the "premium" effect of favorable personality traits occurred irrespective of pre-rated attractiveness [56, 61], we predicted that Paunonen's "honesty premium" effect would be replicated for both relatively attractive and unattractive faces.

### Method

**Ethical statements.** All the procedures of the experiments reported in this article were approved by the Niigata University Ethical Committee on Human Research (2020–0467, 2021–0364). All the experiments were conducted in accordance with the Declaration of Helsinki (2013 amendment) and other relevant guidelines. All participants gave informed consent in advance by marking the "I agree to participate" box on the electronic forms. All the stimulus facial images, including those shown in Fig 1, were adapted from publicly available stock photography (https://www.photo-ac.com/).

**Participants.** Sixty-five undergraduate students (38 women and 27 men; mean age 21.1, *SD* = 2.4) volunteered. They were recruited through electronic flyers distributed after classes and mail-based snowball sampling. The sample size (16 per condition) was determined following the previous study [60].

**Stimulus face images.** Eight grayscale face images (four males, four females, and all East Asians; Fig 1b) with neutral or slightly smiling expressions were adopted from stock photography. Each image showed a face looking about 20–30 years old from the shoulders up. The images were selected by a pilot study in which eight individuals who did not participate in any other experiment observed 32 face images (16 males and 16 females) and rated their physical attractiveness using a 5-point scale (1 = not attractive at all, 5 = extremely attractive). Personality descriptions were not given in the pilot study. Two male and two female images were selected as relatively attractive (mean ratings of 4.2 and 4.3, respectively), and two male and two female images were selected as relatively unattractive (2.7 and 2.9, respectively).

**Personality descriptions.** Each participant read one of four versions of the target descriptions (honesty [low/high] × target gender [M/F]), which were Japanese translations of Paunonen's [60] descriptions. Each description consisted of four sentences describing the target person's intelligence, three sentences describing independence, and two sentences describing honesty. The seven sentences on intelligence and independence served as fillers. We adopted Paunonen's high-intelligence and low-independence descriptions. The last two sentences described the target person as always/never asking permission when borrowing things and having returned/stolen the money from a purse the target person had found (high/low honesty conditions, respectively).

**Procedure.** An electronic questionnaire document (Microsoft Word file) was sent to the participants via e-mail. We created 16 versions of the form in total (two levels of honesty × eight face images [4 male and 4 female]). Each participant was randomly assigned to one of 16 forms. Each form was organized in the following order: a cover page, demographic questions (age, gender), personality description, personality ratings, and physical ratings. Participants were asked to answer one page at a time and not return to a previous page. After completing the demographic questions, they read the personality description of the target person. They then rated seven of the target person's personality traits, as in Paunonen's study [60]: intelligence, independence, honesty, anxiousness, ambition, sociability, and likability. The honesty rating served as a manipulation check. A 9-point rating scale was used (e.g., 1 = extremely dishonest, 9 = extremely honest). No face image was accompanied by the personality description or personality ratings. Following the personality ratings, participants were shown a face image as that of the target person described on the previous page and asked to rate their physical characteristics using a 9-point rating scale. Paunonen's 20 items were used, including attractiveness and filler items (e.g., maturity, eye roundness). S1 Table shows the entire list. Paunonen found an effect of honesty not only on attractiveness but also on seven filler items (e.g., the honest target was judged to have a more feminine face and a more graceful neck).

## Results and discussion

**Data preprocessing.** Missing values (0.85%) were excluded from subsequent analyses. If two adjacent points were marked, we averaged them (e.g., if 2 and 3 were marked, then 2.5).

**Personality ratings.** For each of the seven personality traits, a two-way analysis of variance (ANOVA) was applied to examine the effects of honesty (low/high) and pre-rated facial attractiveness (low/high). S1 Table shows the results, and we selectively report the critical findings here. The manipulation of honesty was successful, as the mean rated honesty was significantly higher in the high honesty condition, $M = 8.1$ ($SD = 1.1$), than in the low honesty condition, $M = 2.7$ ($SD = 2.1$) (main effect of honesty, $F(1, 61) = 165.6$, $p < .001$, $\eta_p^2 = .731$). Furthermore, the high honesty target was rated as more intelligent, more independent, less ambitious, more sociable, and more likable ($ps < .01$). Pre-rated facial attractiveness had almost no effect on personality ratings (see S1 Table) because facial images were not given to the participants at the moment of personality ratings.

**Physical ratings.** The 20 physical characteristics of the target faces were analyzed using the same ANOVA. As can be seen in Fig 2a, rated facial attractiveness was significantly higher for targets with high pre-rated facial attractiveness (main effect of pre-rated attractiveness, $F(1, 61) = 10.71$, $p = .002$, $\eta_p^2 = .149$), confirming the successful manipulation. The high-honesty target faces were rated as more attractive (main effect of honesty, $F(1, 61) = 8.84$, $p = .004$, $\eta_p^2 = .127$). A post hoc power analysis yielded an acceptable statistical power of .856 ($\alpha = .05$). The size of the attractiveness increment by high honesty seemed equal for both attractive and unattractive faces. The two-way interaction was not significant ($F(1, 61) = 0.03$, $p = .852$, $\eta_p^2 = .001$). A Bayes factor analysis showed that a linear model with the two fixed effects (honesty and pre-rated attractiveness) but without the interaction effect was favored against the full model by $BF = 2.96$, which was weak evidence. In addition, the faces of the high honesty targets were rated as kinder ($p < .001$), which was also reported by Paunonen [60]. Honesty did not alter the ratings of the other physical characteristics (see S1 Table). Pre-rated facial attractiveness significantly affected some physical characteristic ratings: attractive faces were rated as having larger and rounder eyes, thinner lips, and more angular shapes. These facial features may (partly) have determined the attractiveness ratings in the pilot study.

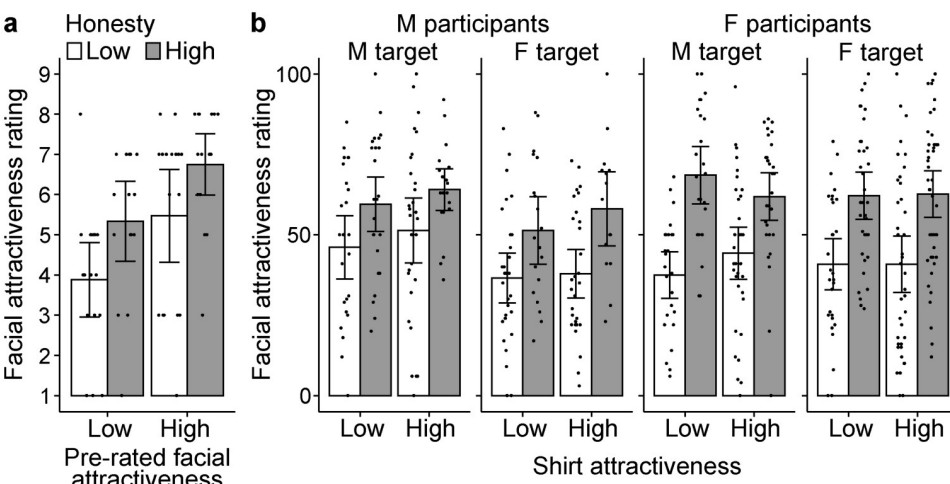

**Fig 2. Results of facial attractiveness rating in Experiments 1 (a) and 2 (b).** Error bars indicate 95% CI of the mean. Each dot indicates one participant's result. M = male, F = female.

**Summary.** Prior presentation of verbal information about honest personality increased the rated attractiveness of subsequently presented target faces. The effect of honesty on facial attractiveness [60] was successfully replicated in a Japanese student sample. This "honesty premium" effect seemed to occur independently of the level of facial attractiveness rated in advance without any personality information.

## Experiment 2

Experiment 2 further tested the replicability of the "honesty premium" effect [60] with a diverse-age sample from a crowdsourcing platform. The larger sample size ($N = 457$) than in Experiment 1 also enabled us to examine the effects of target gender and participant gender, despite Paunonen's [60] report of a lack of a gender effect with $N = 256$. Another purpose of Experiment 2 was exploratory, to test whether attractive clothing influenced the judgments of facial attractiveness. Clothing attractiveness modifies person perception (e.g., [66]), and our previous work [67] found that attractive faces enhance the rated attractiveness of clothing.

Target honesty was manipulated in the same manner as in Experiment 1. We also manipulated the pre-rated attractiveness of the shirts worn by the target persons. A 2 (low/high honesty) × 2 (low/high shirt attractiveness) between-participant design was adopted. Experiment 2 was pre-registered (https://doi.org/10.17605/OSF.IO/NRP58).

### Method

**Participants.** We recruited 480 participants through a crowdsourcing platform (crowdworks.jp). Since we assumed that crowd workers would yield larger variance and more invalid responses than the student sample of Experiment 1, we planned 30 samples per condition. A monetary reward of 300 JPY was given for participation. Data from 23 participants were excluded from the analysis (see below). The remaining 457 participants were 262 women and 195 men, with a mean age of 40.0 ($SD = 10.4$, range 19–75). The experiment was conducted in the Japanese language (thus, most participants were presumed to be residents of Japan, although we did not ask them where they lived).

**Stimulus face images.** We prepared 16 color images showing a person from the waist up by composing face and shirt images using graphic editing software (Fig 1c). We selected 2

male and 2 female faces of medium attractiveness based on the results of the pilot study in Experiment 1. For each gender, two attractive and two unattractive T-shirts were selected (see Fig 1c for examples) based on our previous study [67].

**Procedure.**   We created an online form using lab.js (https://lab.js.org/ [68]). Each participant was randomly assigned to one of eight conditions: honesty (low/high) × shirt attractiveness (low/high) × target gender (M/F). The same personality descriptions were used as in Experiment 1.

The form was organized the same way as in Experiment 1, with the following exceptions: a new set of facial images was used to test the effect of shirt attractiveness. We selected only 8 physical rating items including attractiveness (see S2 Table) from the 20 items used in Experiment 1. All ratings were made on a visual analog scale (VAS) ranging from 0 to 100. The rest of the procedure was virtually identical to that used in Experiment 1.

## Results and discussion

**Data exclusion.**   Data from 22 participants were excluded because of seemingly unserious responses: ratings of 0 (the leftmost of the VAS), 50 (the default value), or 100 (the rightmost) for 8 or more items out of 15. Additionally, we excluded data from one participant who reported a non-binary gender identity because the sample size for the category (i.e., 1) was too small for statistical analysis of the effect of participant gender. Below, we report the results of the remaining 457 participants.

**Personality ratings.**   We conducted a 4-way ANOVA on every rating item to examine the effects of honesty, shirt attractiveness, target gender, and participant gender. Target honesty was manipulated as intended: rated honesty was significantly higher for the high honesty condition, $M = 89.5$, $SD = 13.1$, than for the low honesty condition, $M = 19.1$, $SD = 22.5$ ($F(1, 441)$ = 1490.77, $p < .001$, $\eta_p^2 = .772$). No other main effects or interactions were significant for the honesty ratings ($ps > .1$). Interestingly, the main effect of honesty was significant for all other personality traits as well ($ps < .001$). A high honesty target was rated as more intelligent, more independent, more sociable, more likable, less ambitious, and less anxious. Further details are provided in S2 Table.

**Physical ratings.**   The same 4-way ANOVA was applied to each physical rating item. Attractiveness ratings were key here (Fig 2b): the main effect of honesty was significant ($F(1, 441) = 78.65$, $p < .001$, $\eta_p^2 = .151$), which confirmed the "honesty premium" effect. The achieved statistical power was very high ($\simeq 1$). Shirt attractiveness did not show a significant effect ($F(1, 441) = 1.15$, $p = .285$, $\eta_p^2 = .003$). In addition, male target faces were rated as more attractive than female ones ($F(1, 441) = 6.2$, $p = .013$, $\eta_p^2 = .014$), although we selected face images of equally medium attractiveness. Participant gender had no effect ($F(1, 441) = 0.64$, $p = .425$, $\eta_p^2 = .001$). All interactions were not significant ($ps > .069$). As honesty did not significantly interact with target or participant gender, both male and female participants equally showed the "honesty premium" effect irrespective of target gender.

As shown in S2 Table, honesty showed significant effects on other items as well: the faces of the high honesty target were rated as more feminine, kinder, and healthier ($ps < .05$). Shirt attractiveness significantly influenced only one item: target faces with attractive shirts were rated as healthier ($p = .003$). Participant gender influenced only one item: female participants rated the eyes of the target faces as larger ($p = .020$). Target gender yielded significant effects on five items, including attractiveness, which might reflect physical differences between male and female target images.

**Summary.**   The faces of the target person described as honest were perceived to be more attractive, indicating the presence of the "honesty premium" effect in a diverse age group.

Importantly, this effect occurred irrespective of the participant gender, target gender, and the attractiveness of clothing worn by the target person. Indeed, Bayes factor analyses showed that the linear model with the fixed effect of honesty was more likely compared to the models with honesty×participant gender, honesty×target gender, or honesty×shirt attractiveness (*BF*s = 16.30, 5.01, and 33.24, respectively). We found no evidence that attractive clothing increased the facial attractiveness of the target person.

## Experiment 3

Experiments 1 and 2 replicated the "honesty premium" effect, so we aimed to investigate whether other personality traits have the same effect. Hassin and Trope [57] (Study 6) showed the effect of kindness. Paunonen [60] reported non-significant main effects of intelligence and independence, although there was a somewhat difficult-to-interpret, two-way interaction. These results led us to hypothesize that personality traits that can predict favorable or rewarding interpersonal behaviors would cause a large "premium" effect on facial attractiveness judgments. Indeed, some studies showed that warmth- or moral- related traits (e.g., kindness) often yielded more effect on attractiveness judgments than competence-related traits (e.g., intelligence) [57, 58]. These studies are inspired by the social psychology theory that the human person perception is well described by the two major components of warmth (also referred to as morality or trustworthiness) and competence [69–71].

Experiment 3 examined whether creativity and aggressiveness had an effect on facial attractiveness ratings because aggressiveness is a warmth-related trait while creativity is competence-related. Although it is reported that creativity increases the overall attractiveness of the target person [72], it is still unclear whether creativity influences the physical attractiveness judgments of a target face. We hypothesized that the face of a target person described as low in aggressiveness would be rated more physically attractive than the face of a high aggressiveness target, while the personality descriptions of creativity would not influence the facial attractiveness ratings at all or only slightly.

### Method

**Participants.**   We planned a sample size of 128 under the assumption of medium effect size (f = 0.25) of the main effects and the two-way interaction, with α = 0.05 and power = 0.8. As a result, 117 undergraduates volunteered (82 women and 35 men; mean age 20.5, *SD* = 1.3). They were randomly divided into four groups (low/high creativity × low/high aggressiveness). As in Experiment 1, we recruited the participants via electronic flyers and emails.

**Stimulus face images.**   Four male and four female faces with medium attractiveness were used (see Fig 1d for examples). They were selected based on the results of the pilot study in Experiment 1. As in Experiment 1, the face images were grayscale and showed target persons from the shoulders up.

**Personality descriptions.**   We designed four personality description conditions by manipulating the creativity (low/high) and aggressiveness (low/high) of the target person. As we used both male and female targets, eight variants were prepared. Each description comprised three filler sentences, four sentences on aggressiveness, and four sentences on creativity (See S1 Appendix). For example, a high aggressiveness target was described as often yelling at others and hitting things. A high creativity target was described as often coming up with new ideas and making oil paintings.

**Procedure.**   The experiment was conducted online using Google Forms. We prepared 32 forms (eight face images × four experimental conditions), and each participant was randomly assigned to one of the forms. The forms were organized in the same order as in Experiment 1.

Participants read a personality description of an assigned condition, rated the personality of the target person, saw a stimulus face image, and rated the physical characteristics of the face. We used two personality rating items for manipulation checks (creativity and aggressiveness) in addition to the seven items used in Experiment 1. For physical ratings, we selected 12 items (including attractiveness) from the 20 items used in Experiment 1 (see S3 Table for the entire list). All items were rated using a 9-point scale, in the same manner as in Experiment 1.

## Results and discussion

**Personality ratings.** For each item, the mean ratings were analyzed using a 2-way ANOVA (creativity × aggressiveness). The manipulations were successful. Mean rated creativity was significantly higher for the high creativity condition, $M = 7.8$, $SD = 1.2$, than for the low-creativity condition, $M = 2.6$, $SD = 1.1$ (main effect of creativity, $F(1, 113) = 607.32$, $p < .001$, $\eta_p^2 = .843$), while the main effect of aggressiveness and the 2-way interaction was not significant ($ps > .1$). Mean rated aggressiveness was significantly higher for the high aggressiveness condition, $M = 7.4$, $SD = 1.3$, than for the low aggressiveness condition, $M = 2.0$, $SD = 0.9$ (main effect of aggressiveness, $F(1, 113) = 642.07$, $p < .001$, $\eta_p^2 = .850$), while the main effects of creativity and the 2-way interaction were not significant ($ps > .1$). Many other items were also sensitive to the creativity/aggressiveness manipulation (see S3 Table). On average, a low aggressiveness target was rated as more intelligent, more independent, more honest, less anxious, more sociable, more likable, and less ambitious. A high creativity target was rated as more intelligent, more independent, more honest, more ambitious, and more sociable.

**Physical ratings.** The mean rated physical characteristics were analyzed using the same ANOVA (creativity×aggressiveness). The mean rated attractiveness tended to be higher for faces of low aggressiveness targets ($M = 5.18$, $SD = 1.71$) than for high aggressiveness target faces ($M = 4.58$, $SD = 1.79$), although the statistical significance of aggressiveness main effect was marginal ($F(1, 113) = 3.44$, $p = .066$, $\eta_p^2 = .030$). Creativity did not affect rated attractiveness ($F(1, 113) = 0.17$, $p = .677$, $\eta_p^2 = .002$), nor did the 2-way interaction ($F(1, 113) = 1.70$, $p = .195$, $\eta_p^2 = .015$). The absence of a creativity effect was supported by the moderate Bayes factor favoring the null hypothesis ($BF_{01} = 4.75$). Of the 12 items, we found only one that was significantly modified by the personality descriptions: the face of a high aggressiveness target was rated as less kind (see S3 Table).

**Effect of target gender.** As an ad-hoc analysis, we incorporated target gender into the ANOVA on rated attractiveness because aggression may be perceived differently for male and female aggressors (e.g., [73, 74]). As Fig 3 shows, the high aggressiveness personality description decreased the attractiveness of male faces but not of female faces. A 3-way ANOVA again yielded a marginal main effect of aggressiveness ($p = .072$) and a non-significant main effect of creativity ($p = .628$). The main effect of target gender was not significant ($p = .935$). Importantly, a significant interaction of target gender and aggressiveness was found ($F(1, 109) = 4.81$, $p = .030$, $\eta_p^2 = .042$) and the simple main effect of aggressiveness was significant for male faces ($p = .005$), but not for female faces ($p = .793$). However, the achieved power for this interaction was not high (.615, $\alpha = .05$). No other interactions were significant ($ps > .1$). Although Fig 3 shows a pattern of aggressiveness×creativity interaction for female targets, it was not statistically significant. Future research with a substantial number of participants and target faces may be needed to clarify this issue.

**Summary.** The facial attractiveness of the aggressive male target was rated lower than that of the less aggressive male target. This effect of aggressiveness seemed absent for the female targets. However, such an interaction effect should be examined further with more statistical power. Creativity did not influence rated facial attractiveness.

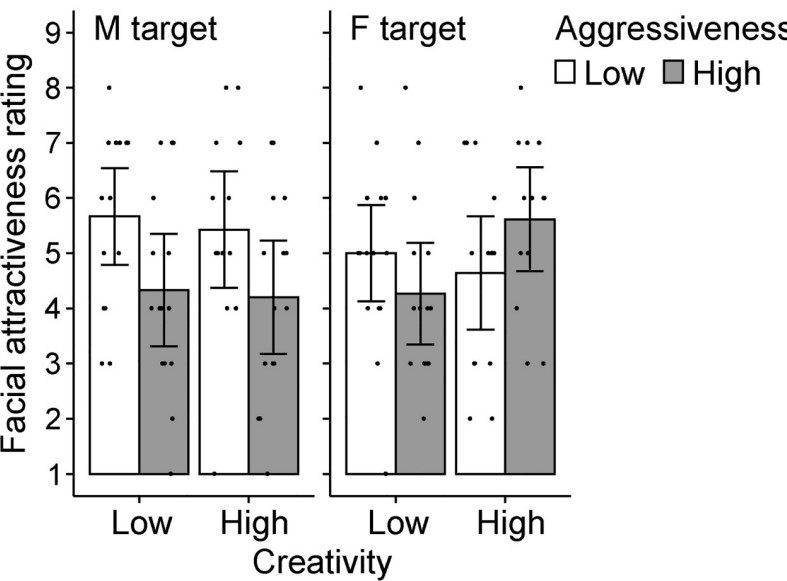

**Fig 3. Results of facial attractiveness rating in Experiment 3.** Error bars indicate 95% CI of the mean. Each dot indicates each participant's result. M = male, F = female.

## Experiment 4

As discussed in the Introduction, Paunonen argued that the perceived personality of a target person affects attractiveness ratings via general likability. Consistent with this view, we observed higher likability ratings and higher facial attractiveness ratings for targets in the high honesty condition (Experiments 1 and 2) and in the low aggressiveness condition (Experiment 3).

Experiment 4 was conducted to examine another potential account for these effects: the participants' mood (see Introduction). We tested whether the honesty descriptions influenced the attractiveness ratings for the face of a different person from the person described as honest/dishonest. We also measured the moods of participants to examine whether the low/high honesty personality description induced a negative/positive mood. Experiment 4 was pre-registered (https://doi.org/10.17605/OSF.IO/PGVA3).

### Method

**Overview.** We designed four conditions based on a personality description of honesty (low/high) and face image relevance (relevant/irrelevant to the personality description). Participants read a personality description, rated the personality of the person described (manipulation check), viewed a face image of the person described (relevant) or a face image of another person (irrelevant), and rated its physical characteristics. Finally, they responded to rating items measuring their own positive and negative moods (Fig 1a). The relevant condition was almost a replication of Paunonen [60] without manipulating intelligence and independence, so we expected that the rated facial attractiveness would be higher for the high honesty condition. In the irrelevant condition, participants were instructed to rate the attractiveness of a face that was irrelevant to the person in the personality description. If a negative/positive mood elicited by reading personality descriptions modified facial attractiveness judgments, the "honesty premium" effect should be observed even for the irrelevant faces.

**Participants.** Eighty-two undergraduate students completed the experiment in exchange for course credit. An additional 49 undergraduates participated and received a voucher reward of 300 JPY. They were recruited through electronic flyers and emails. Because one participant with a monetary reward recognized the personality description we used (see Procedure), we excluded the data of this participant. As a result, we analyzed the results of 130 participants (67 women and 63 men; mean age 19.7, *SD* = 1.1). They were randomly divided into four groups (2 honesty × 2 relevance). A priori estimation of required sample size was 128 under the assumption of medium effect size (f = 0.25) of the two-way interaction (which was the main hypothesis of this experiment) with α = 0.05 and power = 0.8.

**Materials.** Experiment 4 used the same personality descriptions as Experiments 1 and 2. We used the same eight stimulus face images as in Experiment 3 (Fig 1d).

**Procedure.** The experiment was conducted online using Google Forms. We prepared 32 forms (four conditions × eight face images), and each participant was randomly assigned to one of them. Each form comprised a cover page, a personality description, personality ratings, physical ratings, and mood ratings. We adopted the same seven personality rating items as in Experiments 1 and 2. For physical ratings, we used the eight items (including attractiveness) used in Experiment 2. In the relevant condition, the physical ratings page presented a face image and explained that it was the face of the person described on the previous page (Shōta [M] or Misaki [F]). In the irrelevant condition, the page explained that the face image was that of another person (Takumi [M] or Nanami [F]), not the person described on the previous page (Shōta or Misaki). Participants responded to the items using a 9-point scale in the same way as in Experiment 3. Following the physical ratings, participants responded to the mood ratings: "how well do the following words fit your current mood?" We used four positive affect (PA) items (happy, pleased, comfortable, and amusing) and five negative affect (NA) items (depressed, unpleasant, nervous, angry, and anxious). These items were rated on a 7-point scale ranging from 0 (not at all) to 6 (very much). A series of questions followed, asking if the participants recognized any of the personality descriptions or the facial images. These questions were used to find participants who had participated in other experiments reported in this paper.

## Results

**Personality ratings.** As a manipulation check, the mean rated honesty was analyzed using a 2-way ANOVA (honesty×relevance). The honesty manipulation was successful, as confirmed by the significantly higher mean rated honesty for the high honesty condition, *M* = 7.9, *SD* = 1.4, than for the low honesty condition, *M* = 2.2, *SD* = 1.3 ($F(1, 126) = 615.48$, $p < .001$, $\eta_p^2 = .830$). The main effect of relevance and the 2-way interaction were not significant ($F(1, 126) = 1.94$, $p = .166$, $\eta_p^2 = .015$; $F(1, 126) = 0.10$, $p = .753$, $\eta_p^2 = .001$, respectively). As in Experiments 1 and 2, many other personality items were modified by honesty; a high honesty target was rated as more intelligent, more independent, more sociable, more likable, less anxious, and less ambitious (see S4 Table).

**Physical ratings.** Unexpectedly, the effect of honesty on attractiveness ratings was not replicated (Fig 4a). A 2-way ANOVA on attractiveness ratings yielded a non-significant main effect of honesty ($F(1, 126) = 0.90$, $p = .344$, $\eta_p^2 = .007$, $BF_{10} = .016$) and a non-significant interaction with relevance ($F(1, 126) = 0.11$, $p = .737$, $\eta_p^2 = .001$). The main effect of relevance was also not significant ($F(1, 126) = 0.29$, $p = .592$, $\eta_p^2 = .002$). However, we found significant 2-way interactions for two items (mean–kind, stout neck–graceful neck; $ps < .05$) and a marginal interaction for one item (coarse hair–fine hair, $p = .054$; see S4 Table). Importantly, in all of these 3 items, the simple main effect of honesty was significant in the relevant condition but

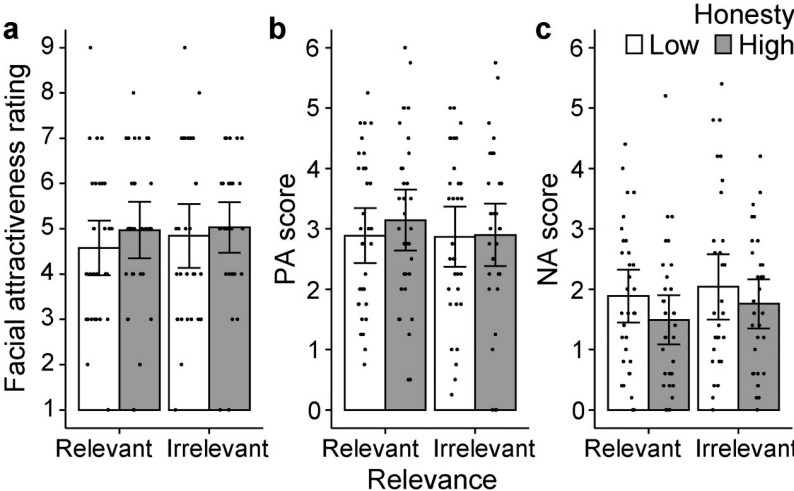

**Fig 4. Results of Experiment 4.** Data of facial attractiveness ratings (**a**) and mood ratings (**b, c**) are shown. Error bars indicate 95% CI of the mean. Each dot indicates each participant's result. PA = positive affect, NA = negative affect.

not in the irrelevant condition, suggesting that the honesty manipulation modified the physical ratings only for the relevant faces. No significant effect was found for any of the other items.

**Mood ratings.** Target honesty did not alter the participants' mood (see Fig 4b and 4c). The same 2-way ANOVAs (honesty × relevance) were applied to PA scores (averages of the four positive affect items) and NA scores (averages of the five negative affect items). The mean PA scores were 3.0 ($SD = 1.4$) for the high honesty condition and 2.9 ($SD = 1.3$) for the low honesty condition, and the main effect of honesty was not significant ($F(1, 126) = 0.35$, $p = .553$, $\eta_p^2 = .003$). Mean NA scores were 1.6 ($SD = 3.0$) and 2.0 ($SD = 2.9$) for high and low honesty conditions, respectively, and the main effect of honesty was not significant ($F(1, 126) = 2.34$, $p = .128$, $\eta_p^2 = .018$). The main effect of relevance and its interaction with honesty were not significant for either PA or NA scores ($ps > .1$). When the same ANOVA was applied to each item, we found only one NA item ("unpleasant") to be modified significantly by honesty ($F(1, 126) = 6.19$, $p = .014$, $\eta_p^2 = .047$); reading the low-honesty target description induced an unpleasant mood.

The PA and NA scores did not relate to the facial attractiveness ratings. A multiple linear regression model to fit the rated attractiveness with the independent variables of PA and NA scores found that neither of the regression coefficients was statistically significant ($b = 0.24$ and 0.11, respectively; $ps > .05$). Against this model, the null model was more likely ($BF_{01} = 3.67$). Likewise, the NA item "unpleasant" did not account for the attractiveness ratings ($b = -0.03$, $p = .774$; $BF_{01} = 5.14$).

## Discussion

Since the "honesty premium" effect was not replicated, even in the relevant condition, we could not determine whether the effect was attributable to participants' moods. The replication failure might be due to the use of course credit. The students who participated in exchange for course credit were assigned a short report on the experiment, and they might have had a critical perspective on the content of the experiment. Indeed, the effect size of honesty on the rated facial attractiveness in the relevant condition (high honesty condition minus low honesty condition) was smaller (0.24) for the participants who received course credit than for those who received a monetary reward (0.66).

Despite replication failure, the pattern of the results was generally inconsistent with the mood hypothesis. First, the participants' mood ratings did not correlate with the facial attractiveness ratings. Second, a significant effect of honesty on the physical ratings of target faces was found only in the relevant condition. Nevertheless, we could not draw definite conclusions on the mood hypothesis because of the replication failure of the honesty premium effect. Further study is warranted.

## General discussion

We successfully replicated Paunonen's [60] honesty premium effect in Experiments 1 and 2 (but not in Experiment 4). The effect was independent of the pre-rated facial attractiveness of target person, participant gender, target gender, and target clothing attractiveness. The effect of honesty was found not only for facial attractiveness but also for other physical features. These findings are largely consistent with those of previous studies [56–58, 60–62], indicating that the effect is robust and reproducible. We also found that the information of a less aggressive personality might increase the facial attractiveness of a male target (Experiment 3), and that of a creative personality did not have any effect on facial characteristic ratings. Together with the previous reports, the information of high honesty, high kindness, and low aggressiveness (of a male target) may enhance the facial attractiveness of the target, whereas intelligence, independence, and creativity did not (although Paunonen [60] reported an interaction effect of intelligence and independence).

These findings indicate that not all desirable personality traits increase the facial attractiveness ratings. Why do some traits modify the perception of facial attractiveness and other traits do not? As the classical experiment of Asch [75] showed, some "central" personality traits may have critical roles in person perception compared to "peripheral" traits. The perception of warmth-related personality traits, such as honesty, may raise the perceived attractiveness of a target face as those traits predict favorable and rewarding interpersonal behaviors. This is consistent with the view that facial attractiveness is utilized as a criterion that leads observers to approach/avoidance behaviors. As noted in Experiment 3, the recent theories of social perception claim that the structure of the perception of others has two main factors, warmth and competence [69–71]. However, the perception of competence-related traits seems to have little or no effect on the perception of facial attractiveness.

Facial attractiveness is a psychological composite of various sources of information. This means that it is sensitive to non-physical information and therefore malleable. The visual mechanism tuned for some "attractive" facial features, such as symmetry and averageness, should be considered as a component in relation to other components that make up the psychological system of attractiveness perception.

### The mechanism by which perceived personality modifies facial attractiveness

Paunonen [60] concluded that increased facial attractiveness was mediated by general likability, and our results were consistent with this view. The rated likability of a target person was increased by honest personality descriptions (Experiments 1 and 2). The mediation effect of likability was further confirmed by a mediation analysis on Experiment 2, which had a large sample. A significant direct effect from the rated honesty to the rated attractiveness ($b = 0.26$, $p < .001$) became non-significant ($b = -0.04$, $p = .63$) by controlling the rated likability as a mediator. The indirect effect mediated by the rated likability was statistically significant (Sobel's test, $z = 4.35$, $p < .001$; see also S1 Fig). In Experiment 3, we found that the rated likability was significantly lower for the male target (3.9) than for the female target (5.4) in the

high aggressiveness condition but there was no gender effect in the low aggressiveness condition (7.2 and 6.8, respectively), which was consistent with the interaction effect found for rated facial attractiveness in Experiment 3. The only exception was Experiment 4, in which honesty manipulation influenced likability but not facial attractiveness.

Experiment 4 tested an alternative account: whether a positive/negative mood elicited by the perception of an honest/dishonest personality would increase/decrease facial attractiveness. We did not find evidence that the participants' mood correlated with the rated attractiveness. Although Experiment 4 failed to replicate the honesty effect on attractiveness ratings, it was noteworthy that the face of a high-honesty target was rated as kinder, having a more graceful neck and finer hair in the relevant condition, while there was no effect of honesty on physical ratings in the irrelevant condition. These results imply that a mediating effect of mood is relatively unlikely.

### Limitations and implications for future research

In Experiment 4, we failed to replicate the effect of honesty on facial attractiveness. Experiment 4 also suggests that a high likability of the target does not necessarily increase facial attractiveness. The psychological mechanism behind the "good is beautiful" effect should be examined in further detail.

Another issue to be considered is the critical nature of personality traits that influence facial attractiveness judgments. We presumed that personality traits that could predict favorable or rewarding interpersonal behaviors, such as high honesty and low aggressiveness, would increase facial attractiveness. Perception of these traits would also increase likability, which is consistent with the mechanism proposed by Paunonen [60]. We may test this hypothesis through further experiments that manipulate participants' expectations of rewarding interpersonal behaviors.

Finally, the ecological validity of our experiments is limited. The generality of the "good is beautiful" effect in more ecologically valid situations should be investigated in future research.

### Conclusions

The perception of an honest personality of male and female target persons enhanced the target facial attractiveness rated by male and female participants. It was also implied that the perception of a less aggressive personality of a male target increased facial attractiveness. Creative personality did not yield such effect. Perceptions of warmth-related traits through verbal information may alter the perceived facial attractiveness of the target person, whereas competence-related traits seem to have only partial or little effect. The "honesty premium" effect on facial attractiveness was replicated with a diverse-age sample and was observed independent of target gender, participant gender, target clothing, and pre-rated target facial attractiveness. Paunonen [60] suggested that this effect is mediated by general likability, and our results were largely consistent with this account. We did not find evidence for the hypothesis that a positive mood induced by the perception of an honest personality would enhance facial attractiveness.

These results suggested the presence of a "what is good is beautiful" stereotype in addition to the physical attractiveness stereotype ("what is beautiful is good"). Facial attractiveness is formed not only by physical features but is a psychological composite of multiple factors made by integrating a wide range of information related to the target person.

### Supporting information

**S1 Fig. Mediation analysis on Experiment 2.**
(PDF)

**S1 Table. Results of Experiment 1.**
(PDF)

**S2 Table. Results of Experiment 2.**
(PDF)

**S3 Table. Results of Experiment 3.**
(PDF)

**S4 Table. Results of Experiment 4.**
(PDF)

**S1 Appendix. Experiment 3 personality descriptions.**
(PDF)

## Author Contributions

**Conceptualization:** Ryosuke Niimi, Mami Goto.

**Data curation:** Ryosuke Niimi, Mami Goto.

**Formal analysis:** Ryosuke Niimi, Mami Goto.

**Funding acquisition:** Ryosuke Niimi.

**Investigation:** Ryosuke Niimi, Mami Goto.

**Methodology:** Ryosuke Niimi, Mami Goto.

**Project administration:** Ryosuke Niimi.

**Supervision:** Ryosuke Niimi.

**Visualization:** Ryosuke Niimi.

**Writing – original draft:** Ryosuke Niimi.

**Writing – review & editing:** Ryosuke Niimi.

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
