## [Decision Letter · Decision Letter 0]

9 Nov 2022

PONE-D-22-24839Good conduct makes your face attractive: The effect of personality perception on facial attractiveness judgmentsPLOS ONE

Dear Dr. Niimi,

Thank you for submitting your manuscript to PLOS ONE. After careful consideration, we feel that it has merit but does not fully meet PLOS ONE’s publication criteria as it currently stands. Therefore, we invite you to submit a revised version of the manuscript that addresses the points raised during the review process.

Both Reviewers raised several issues that need to be addressed (e.g., results from experiment four and various other statements). Thus, I invite the authors to review their manuscript strictly following the Reviewer's statements and then resubmit their revised manuscript.

We look forward to receiving your revised manuscript.

Kind regards,

Vilfredo De Pascalis

Academic Editor

PLOS ONE

Journal Requirements:

"RN received a grant from the KOSE Kosmetology Research Foundation. MG declares no competing interests."

5. We note that Figure 1 includes images of a participants in the study. 

Additional Editor Comments:

Both Reviewers raised several issues that need to be addressed (e.g., results from experiment four and various other statements).

Thus I invite the authors to review their manuscript strictly following the Reviewer's statements and then resubmit their revised manuscript.

Reviewers' comments:

Reviewer's Responses to Questions

**Comments to the Author**

1. Is the manuscript technically sound, and do the data support the conclusions?

Reviewer #1: Partly

Reviewer #2: Yes

2. Has the statistical analysis been performed appropriately and rigorously? 

Reviewer #1: No

Reviewer #2: Yes

3. Have the authors made all data underlying the findings in their manuscript fully available?

Reviewer #1: Yes

Reviewer #2: Yes

4. Is the manuscript presented in an intelligible fashion and written in standard English?

Reviewer #1: Yes

Reviewer #2: Yes

5. Review Comments to the Author

Reviewer #1: The manuscript was generally well written. I note a few confusing statements below. My greatest concern is how Experiment 4 produces inconclusive results but the authors interpret them to mean that positive or negative affect is not involved in the process. I believe the contribution would be more scientifically clear if Exp. 4 were removed or if the researchers conducted another experiment to clarify the results of Exp. 4. Below I give my comments in the order in which they appeared in the manuscript.

In the abstract, the authors write "more or less concerned." This is unclear. Writing, "Most people are at least somewhat concerned..." would be more clear.

In the abstract, the authors write "personal traits" when they mean "personality traits" (I believe).

In the abstract, the word "influence" is used, but it would be more accurate to write "correlated." This is because "influence" implies causality but most of the research is correlational (it is difficult to randomly assign folks to physical attractiveness levels).

Page 4, line 17 -- "yielded" is unclear; "gave" would be more accurate.

In Study 1, the authors explain the sample size by writing that it was determined following the previous study. This is not clear. Simply using the same sample size as past is not good practice. Power calculations are needed. This same comment applies to all experiments.

Page 7, line 5 -- "with the appearance of 20s" is unclear. Do you mean that the photo was shown for 20s? Just state it more directly, if so.

Page 8, line 27 -- "seemed equal" -- At several places in the results, the authors want to claim that they proved the null hypothesis. This is incorrect. Bayes or some other statistics are needed to do this. See Aczel, B., Palfi, B., Szollosi, A., Kovacs, M., Szaszi, B., Szecsi, P., ... & Wagenmakers, E. J. (2018). Quantifying support for the null hypothesis in psychology: An empirical investigation. Advances in Methods and Practices in Psychological Science, 1(3), 357-366.). To claim support for a null hypothesis, Bayes statistics need to be used (jamovi and JASP are free software that can easily be used to calculate Bayes factors).

Page 10, line 26-27: it is not clear why the sample size was too small to examine participant gender. This is also confusing because gender is tested in the middle of page 11.

Experiment 3 nicely reported a priori sample size determination methods.

In Exp. 3, target creativity and aggressiveness were crossed in the manipulation, so that a target could be high creative and high aggressive. Why was this done? I suspect it complicates the results and probably led to the weaker results and lack of an effect in female targets (the odd, almost interactive effect in female targets suggests this). This should be discussed.

Report standard deviations with each mean.

In Exp. 4, the manipulation of relevance is a good way to test the mood explanation. The measurement of mood is not. The mood ratings were made by participants after all other measures, rendering an insensitive measure even more insensitive. Also, the instructions given to participants for the mood ratings are not provided. They should be.

What happens when you test each mood word, as was done for the attractiveness ratings? The NA results are close to significant.

In the end, Exp. 4 failed to replicate the honesty effect and the experiment does not aid in interpreting the effects. There are many possible reasons for the failure to replicate.

In the discussion, mediation (via likability) is mentioned, but no formal statistical mediation is tested. Inclusion of these analyses would benefit the interpretation of the results.

Reviewer #2: Review of “Good Conduct Makes Your Face Attractive: The Effect of Personality Perception on Facial Attractiveness Judgments”

PONE-D-22-24839

Reviewed for PLOS ONE

(November, 2022)

Although I did not find this an exceptionally strong or groundbreaking set of studies, I also didn’t have a strong reason why the paper should be rejected from this journal. I mostly had suggestions for further clarification in various statements made in the manuscript, as listed below.

Comments (issues that need clarification):

p. 3; line 27: What do you mean by “how to deal with physical attractiveness stereotypes”?

p. 3; line 28: I would add, “Perceptions of..” or “Judgements of…” before Facial attractiveness.

p. 3; line 29: What is the meaning of “shared tastes”?

p. 3; line 30: I would change to “factors affecting perceptions of facial attractiveness…”

p. 4; line 2: Could you say more about the meaning of “hair,” such as length or color?

p. 4; line 3: You might change to “Red color was first shown to increase…”

p. 4; line 9: I would add “the target’s” before facial attractiveness.

p. 4; line 10: What is the meaning of a “unfairly disadvantaged target person (and “unfairly advantaged target” later in the sentence)? Also, I would add “a” and “an” before these phrases.

p. 4; line 12: Can you give an example in parentheses of “Favorable behaviors”?

As you discuss other factors that may influence perceptions of physical attractiveness, you might also refer to prior interactions with the person, as I believe there is research to show that if there is prior interaction (even very limited), the ratings are higher.

p. 5; line 4: You might not want to refer to a “difficult-to-interpret interaction effect” unless you are to explain it.

p. 5; line 10: Can you explain more about ear length and chin shape?

p. 5; line 16: For Experiment 2, you might also refer to the geographical location here, since you do for Experiment 1.

p. 5 line 23: I find it a little odd that shirt attractiveness was varied in an experiment, and would like to see more justification for doing so.

p. 6; I’m a little confused about the pre-registered study on the issue of mood, and wondered why more information about this study was not presented in the manuscript.

p. 7; line 5: You might refer to it as a pilot study rather than a preliminary survey.

p. 7; line 8: You might add “in the pilot study” after “Personality descriptions were not given.”

p. 8; top: When you refer to the 20 items from Paunonen including attractiveness, you might give a couple of examples here in text.

p. 8; lines 4-6: I found this sentence (that begins, “Of interest was…”) to be confusing.

p. 8; line 11: How was it possible in an online survey that two adjacent points could be marked? Or was it a paper survey?

p. 8; bottom: Aren’t the findings related to the effect of pre-rated facial attractiveness on facial attractiveness just a manipulation check?

p. 12: I found it a little odd that you added creativity this study with the anticipation of not finding an effect. Why not include a variable for which you expected a significant effect?

p. 15; line 13 and line 17: I am confused about the meaning of relevant/irrelevant, and think more information needs to be presented.

p. 16; line 4: I am confused by this information. You state that you prepared 32 forms, and each participant was randomly assigned to one of them. But you had only 130 participants, and thus, how could you make valid group comparisons? If this truly is the case, you may need to consider adding more participants or even eliminating this study. Could this also be why the honesty effect was not replicated?

Experiment 4: If mood really was of interest, why wasn’t mood a manipulated independent variable?

General Discussion: It would seem that you should discuss honesty as a “central trait” and the social psych research on how central traits can change the interpretation of other information presented simultaneously.

Very minor editing suggestions:

p. 3; line 11: I would add “a” before stereotype.

p. 9; line 17: I would add “a” before “crowdsourcing platform”

p. 11; line 28: Add “the” before “target person”

p. 14; line 29: Add “the” before perceived personality” and add “a” before target person.

p. 15; line 4: Add “the” before participants.

.

6. PLOS authors have the option to publish the peer review history of their article (what does this mean?). If published, this will include your full peer review and any attached files.

Reviewer #1: No

Reviewer #2: No

---

## [Author Response · Author response to Decision Letter 0]

9 Dec 2022

(Please see Response To Reviewers file)

---

## [Decision Letter · Decision Letter 1]

26 Jan 2023

PONE-D-22-24839R1Good conduct makes your face attractive: The effect of personality perception on facial attractiveness judgmentsPLOS ONE

Dear Dr. Niimi,

Thank you for submitting your manuscript to PLOS ONE. After careful consideration, we feel that it has merit but does not fully meet PLOS ONE’s publication criteria as it currently stands. Therefore, we invite you to submit a revised version of the manuscript that addresses the points raised during the review process.

We look forward to receiving your revised manuscript.

Kind regards,

Vilfredo De Pascalis

Academic Editor

PLOS ONE

Journal Requirements:

Additional Editor Comments:

I see that the authors addressed all the suggested reasonable changes in the first step of revision. I am sorry that only one revision was available in the second step. However, the second Reviewer was satisfied with the addressed changes, although he also identified several minor issues that should be clarified in the manuscript. 

Given that the time is going on and assuming that the suggested changes are quickly addressed, I invite the author to make all the proposed minor changes and submit them directly to my attention for acceptance.

Reviewers' comments:

Reviewer's Responses to Questions

**Comments to the Author**

1. If the authors have adequately addressed your comments raised in a previous round of review and you feel that this manuscript is now acceptable for publication, you may indicate that here to bypass the “Comments to the Author” section, enter your conflict of interest statement in the “Confidential to Editor” section, and submit your "Accept" recommendation.

Reviewer #2: (No Response)

2. Is the manuscript technically sound, and do the data support the conclusions?

Reviewer #2: Yes

3. Has the statistical analysis been performed appropriately and rigorously? 

Reviewer #2: Yes

4. Have the authors made all data underlying the findings in their manuscript fully available?

Reviewer #2: Yes

5. Is the manuscript presented in an intelligible fashion and written in standard English?

Reviewer #2: Yes

6. Review Comments to the Author

Reviewer #2: Re-review of “Good Conduct Makes your Face Attractive: The Effect of Personality Perception on Facial Attractiveness Judgments”

Reviewed for PLOS ONE

The authors have revised the manuscript in a satisfactory way and also written a reasonable cover letter indicating how they have responded to reviewer suggestions from the first round of review. Below, I identify several issues that could still use clarification in the manuscript.

Abstract; line 3: You state that human facial attractiveness is related to physical features, such as averageness and symmetry. I think some readers will have no idea of what you mean by averageness, as it can have another meaning too in regard to physical attractiveness (just being an average looking person). I would suggest you provide a different example, such as “clear complexion.”

Abstract; line 5: When you write, “Here, we examined the effect…” add “judgments of attractiveness”.

Abstract; line 11: The phrase, “whereas aggressive personality of a male target might decrease the rated facial attractiveness” should be replaced with what finding you actually obtained.

p. 3; line 12: I found it confusing (and a little annoying actually) when you have a sentence such as “It is noteworthy that the validity of such a stereotype has been questioned (but see…). First, you don’t say how the validity has been questioned, and then you don’t give us an idea of what the “but see” would say. So, both parts of the sentence seem misleading, and particularly in combination.

p. 3; line 20; ditto the two “but see”.

p. 3; line 24 and 25: Judgments of attractiveness are not the same as “being attracted”, but this sentence implies they are the same.

p. 3; line 25: You state that attractiveness is not an objective reality and that beauty is in the eye of the beholder. However, earlier in the same page (line 17), you had indicated that visual perception of facial attractiveness is consistent among observers.

p. 3; line 28: Can you provide a citation at the end of this sentence saying that appearance-related prejudice and stereotypes may undermine psychological wellbeing?

p. 3; line 30: I’m still not sure what you mean by “private tastes”. I think you could choose a different wording.

p. 4; line 13: I think you give enough other examples that it would be fine to delete the example of the study referring to an “unfairly disadvantaged job applicant” and an “unfairly advantaged applicant”. I think that is just a confusing example.

p. 5; line 5: Maybe change to: a personality description that included the manipulations of honesty, intelligence, and independence.

p. 7; top; Explain more where the participants were obtained from and how they were recruited.

p. 7: Because you had both male and female targets (and described that at the top of p. 7), it is confusing when you don’t refer to the gender of the target at the bottom of the page (that it is just ignored).

p. 8; line 24: You state that “Pre-rated facial attractiveness had almost no effect…” That sentence seems too ambiguous. Either there are effects or there are none.

p. 9; Summary: In this brief Summary, summarize again the findings that lead to the conclusion that there was an “honesty premium” effect.

p. 9; bottom: I’m still confused why clothing was manipulated rather than facial attractiveness.

p. 10; line 19: Provide another statement about what the two attractiveness t-shirts looked like and what the unattractive t-shirts looked like.

p. 12; top: Were there any significant interactions?

p. 12; line 20: You might consider adding this phrase in parentheses after “attractiveness ratings because aggressiveness (or its opposite)…

p. 13; Participants: Where did the participants come from and how were they recruited?

p. 16: Ditto the question for Experiment 4.

7. PLOS authors have the option to publish the peer review history of their article (what does this mean?). If published, this will include your full peer review and any attached files.

Reviewer #2: No

---

## [Author Response · Author response to Decision Letter 1]

29 Jan 2023

Please see the attached file "ResponseToReviewers.docx".

---

## [Editor Report · Decision Letter 2]

1 Feb 2023

Good conduct makes your face attractive: The effect of personality perception on facial attractiveness judgments

PONE-D-22-24839R2

Dear Dr. Niimi,

We’re pleased to inform you that your manuscript has been judged scientifically suitable for publication and will be formally accepted for publication once it meets all outstanding technical requirements.

Kind regards,

Vilfredo De Pascalis

Academic Editor

PLOS ONE

Additional Editor Comments (optional):

The authors have addressed all the minor comments of the Reviewer#2. Thus I think can now be accepted for publication.
---

## [Editor Report · Acceptance letter]

3 Feb 2023

PONE-D-22-24839R2 

Good conduct makes your face attractive: The effect of personality perception on facial attractiveness judgments 

Dear Dr. Niimi:

I'm pleased to inform you that your manuscript has been deemed suitable for publication in PLOS ONE. Congratulations! Your manuscript is now with our production department. 

Kind regards, 

on behalf of

Prof. Vilfredo De Pascalis 

Academic Editor

PLOS ONE